# Anti-Osteoporotic Effects of Kukoamine B Isolated from *Lycii Radicis* Cortex Extract on Osteoblast and Osteoclast Cells and Ovariectomized Osteoporosis Model Mice

**DOI:** 10.3390/ijms20112784

**Published:** 2019-06-06

**Authors:** Eunkuk Park, Jeonghyun Kim, Mun-Chang Kim, Subin Yeo, Jieun Kim, Seulbi Park, Miran Jo, Chun Whan Choi, Hyun-Seok Jin, Sang Woo Lee, Wan Yi Li, Ji-Won Lee, Jin-Hyok Park, Dam Huh, Seon-Yong Jeong

**Affiliations:** 1Department of Medical Genetics, Ajou University School of Medicine, Suwon 16499, Korea; jude0815@hotmail.com (E.P.); danbi37kjh@hanmail.net (J.K.); monotos@hanmail.net (M.-C.K.); snsnans@naver.com (S.Y.); eeseul0707@naver.com (J.K.); seulbi@ajou.ac.kr (S.P.); jjo9313@naver.com (M.J.); 2Department of Biomedical Sciences, Ajou University Graduate School of Medicine, Suwon 16499, Korea; 3Natural Products Research Institute, Gyeonggi Institute of Science & Technology Promotion, Suwon 16229, Korea; cwchoi78@gmail.com; 4Department of Biomedical Laboratory Science, College of Life and Health Sciences, Hoseo University, Asan 31499, Korea; microchin@hanmail.net; 5International Biological Material Research Center, Korea Research Institute of Bioscience and Biotechnology, Daejeon 34141, Korea; ethnolee@hanmail.net; 6Institute of Medicinal Plants, Yunnan Academy of Agricultural Sciences, Kunming 650200, China; wyli2012@126.com; 7Korea Food Research Institute, Seongnam 13539, Korea; dnjs0004@naver.com; 8Dongwoodang Pharmacy Co. Ltd. Yeongchen 38819, Korea; navy9376@hanmail.net

**Keywords:** osteoporosis, herbal medicine, kukoamine B, osteoblast, osteoclast, bone mineral density, ovariectomized mice

## Abstract

Osteoporosis is an abnormal bone remodeling condition characterized by decreased bone density, which leads to high risks of fracture. Previous study has demonstrated that *Lycii Radicis* Cortex (LRC) extract inhibits bone loss in ovariectomized (OVX) mice by enhancing osteoblast differentiation. A bioactive compound, kukoamine B (KB), was identified from fractionation of an LRC extract as a candidate component responsible for an anti-osteoporotic effect. This study investigated the anti-osteoporotic effects of KB using in vitro and in vivo osteoporosis models. KB treatment significantly increased the osteoblastic differentiation and mineralized nodule formation of osteoblastic MC3T3-E1 cells, while it significantly decreased the osteoclast differentiation of primary-cultured monocytes derived from mouse bone marrow. The effects of KB on osteoblastic and osteoclastic differentiations under more physiological conditions were also examined. In the co-culture of MC3T3-E1 cells and monocytes, KB promoted osteoblast differentiation but did not affect osteoclast differentiation. In vivo experiments revealed that KB significantly inhibited OVX-induced bone mineral density loss and restored the impaired bone structural properties in osteoporosis model mice. These results suggest that KB may be a potential therapeutic candidate for the treatment of osteoporosis.

## 1. Introduction

Bone is a living organ constantly remodeled and maintained by a balance between bone formation and resorption [1,2]. Bone remodeling is a physiological condition regulated by osteoblasts replacing new bone formations and osteoclasts removing old or damaged bone [1]. Osteoblasts, which are differentiated from mesenchymal stem cells, play an important role in bone formation by synthesizing various bone matrix proteins and transporting minerals into the matrix. Osteoclasts are differentiated from mononuclear cells of the monocyte/macrophage lineage, and they break down bone tissue essential for osteoblast new bone formation. Bone formation includes the proliferation and differentiation of osteoblasts via the activation of alkaline phosphatase (ALP), collagen synthesis, and the mineralization of bone [3]. Bone resorption involves the differentiation of osteoclasts via the activation of tartrate-resistant acid phosphatase (TRAP) [4,5]. The balanced relationship between bone formation and bone resorption is critical for maintaining bone strength and preventing bone loss [6]. However, an imbalanced regulation of these bone-remodeling processes leads to serious bone loss via osteoporosis, a metabolic bone disease [1]. 

Osteoporosis is a serious condition that causes a loss of bone density and strength, leading to increased, painful bone fractures [7]. Osteoporotic fractures are a serious health problem, particularly for aging, postmenopausal women [8]. Currently, various pharmacological treatments for osteoporosis are proven to reduce fracture risks, including bisphosphonates, which inhibit bone resorption by osteoclasts; selective estrogen receptor modulators; the monoclonal antibody to the receptor activator of nuclear factor kappa-B ligand (RANKL); and a parathyroid hormone [9]. 

Many herbal medicines derived from natural products have been used in the treatment of osteoporosis, with fewer negative effects [10,11,12,13,14,15]. In addition, bioactive components and related properties for alternative treatments of osteoporosis have been identified [16,17,18,19]. The plant *Lycii Radicis* Cortex (LRC) has been used in traditional medicines in East Asia. Physiological bioactive properties, such as apigenin, luteolin, kaempferol, quercetin, oleanolic acid, dihydrophaseic acid, and urosolic acid, were identified in LRC extracts [20,21]. The authors’ previous study suggested that LRC extract promoted osteoblast differentiation and inhibited the loss of bone mineral density (BMD) in an osteoporotic mice model, without negative side effects [22]. Furthermore, LRC prevented osteoclast differentiation induced by RANKL through the down-regulation of osteoclastogenesis-related markers [23]. Recently, dihydrophaseic acid 3′-*O*-β-d-glucopyranoside was identified as a candidate bioactive compound for the anti-osteoporotic effects of LRC extract [24]. However, because only a small amount of this compound is contained in LRC extract, other component(s) responsible for the bone formation-enhancing effect of LRC may still be unidentified.

This study aimed to identify the other bioactive component(s) for anti-osteoporotic effects in LRC extract. First, a fractionation of the ethanol LRC extract was performed, and then a single compound was isolated. Next, the anti-osteoporotic effects of the compound with in vitro and in vivo osteoporosis models were investigated.

## 2. Results and Discussion

### 2.1. A Bioactive Component Promoting Osteoblast Differentiation Was Isolated from LRC Extract

The authors’ previous study reported that the ethanol extract of LRC induced osteoblast differentiation in preosteoblast MC3T3-E1 cells, and prevented the loss of bone mineral density in ovariectomized (OVX) mice [22]. The study identified the 13 main constituents using a high-performance liquid chromatography (HPLC)–electrospray ionization (ESI)–tandem mass spectrometry system [22]. In the present study, fractionation and isolation of the bioactive component from LRC ethanol was performed (Appendix A). 30% of the ethanol LRC extract was fractionated, and each fraction was investigated for a bone formation-enhancing effect using an alkaline phosphatase (ALP) activity assay in preosteoblast MC3T3-E1 cells for identifying the bioactive fraction (Appendix A). ALP is a homodimeric protein enzyme located in the cell membranes of osteoblasts, and is a reliable marker of bone metabolism during osteoblast differentiation [25]. Therefore, an ALP assay was conducted to screen the bioactive fractions on bone formation. Each fraction was treated in an osteoblastic cell line for three days, and osteoblast differentiation was measured by the ALP assay. Consequently, a bioactive single compound, kukoamine B (KB), was isolated from the LRC extract and identified by nuclear magnetic resonance (NMR) and mass spectrometry analyses (Figure 1 and Appendix A). The composition of KB in the LRC extract was 0.657%.

KB is a natural spermine alkaloid compound that was first isolated from LRC [26]. It has been reported that KB neutralizes both lipopolysaccharides (LPS) and oligodeoxynucleotides containing CpG motifs (CpG DNA), and inhibits LPS- or CpG DNA-mediated proinflammatory signal transduction and cytokine expression [27,28]. The biological characteristics of KB on osteoporosis have not been studied.

### 2.2. KB Increased Osteoblast Differentiation and the Mineralized Nodule Formation of Preosteoblastic MC3T3-E1 Cells

First, the effects of KB on osteoblast differentiation in the MC3T3-E1 cell line were examined. Osteoblast differentiation is characterized by three stages: (1) Cell proliferation, (2) matrix maturation, and (3) matrix mineralization [29]. To evaluate the effect of KB on osteoblast differentiation, orthodox methods were used, including ALP activity, cell viability, and mineralization assays. ALP activity plays a crucial role in the mineralization of newly-formed bone and increases during osteoblast differentiation [30,31]. Cells were cultured with three different concentrations of KB (5, 10, and 20 μM), and ALP activity was assessed at two, three, four, and five days (Appendix A). ALP activities in the KB-treated cells (5 and 10 μM) were increased at three and four days, but then decreased at day five (Appendix A). After incubating for five days with KB (5, 10, and 20 μM), cell viability was analyzed using a water-soluble tetrazolium salt (WST) assay. A higher ALP activity was observed for the treatments of 10 and 20 μM KB, with increased ALP-positive staining colonies in osteoblastic MC3T3-E1 cells compared to the control group (Figure 2A,C). Treatments of KB of 5, 10, and 20 μM did not affect the cell viability of osteoclastic cells (Figure 2B).

Next, the effects of KB on mineralized nodule formation in MC3T3-E1 cells were examined. Bone matrix is mineralized by osteoblast differentiation, leading to the induction of calcium and phosphate-based minerals. Consequently, bone mineralization develops with several matrix proteins [32]. Many studies have suggested that the mineralized osteoblast is a characteristic method for testing the effects of drug treatments on calcium deposition and bone formation [33,34]. Alizarin red S is a histochemical method commonly used for evaluating calcium-rich deposits in the mineralization of osteoblast cells [35]. Calcium phosphate and osteoblast mineralization with positive alizarin red S staining revealed successful mineralized osteoblast cells in vitro [36]. The MC3T3-E1 cells were treated with osteoblast induction reagents (ascorbic acid and β-glycerophosphate) for 21 days, resulting in mineralized nodule formation and the osteoblast mineralization of MC3T3-El cells. Co-treatment of KB (10 and 20 μM) with induction reagents presented higher positive alizarin red S staining colonies than for the non-treated control cells. These results demonstrated that KB enhanced the differentiation and mineralized nodule formation of bone-forming osteoblasts.

### 2.3. KB Increased mRNA Expression of Osteoblastic Differentiation-Related Genes

The augmentation of osteoblast differentiation is closely related to high expressions of the key osteoblastic marker genes *Alpl* (alkaline phosphatase, tissue-nonspecific isozyme, ALP), *Bglap* (bone gamma-carboxyglutamic acid-producing protein, Osteocalcin), and *Sp7* (Osterix) [30]. ALP is a sensitive and reliable indicator of bone metabolism. Its expression and activity increase during bone formation activity at the site of mineral deposition, indicating that ALP is an important enzyme in the mineralization of newly-formed bone [31]. Mature osteoblasts express high levels of osteocalcin, which is associated with bone mineralization and calcium ion homeostasis [37]. The Sp7 transcription factor regulates bone-forming osteoblasts [38].

To further confirm the effects of KB on the cellular differentiation of osteoblasts, the expression of osteoblastic markers, including *Alpl*, *Bglap*, and *Sp7*, was investigated. Osteoblastic MC3T3-E1 cell lines were treated with 20 μM of KB. After treatment with KB, the mRNA expressions of *Alpl*, *Bglap*, and *Sp7* were measured by quantitative reverse-transcription polymerase chain reaction (qRT-PCR). Significantly increased expressions of *Alpl*, *Bglap*, and *Sp7* were observed in KB-treated cells compared to the control (Figure 3). These results suggested that KB promoted the up-regulation of the mRNA expressions of *Alpl*, *Bglap*, and *Sp7*.

### 2.4. KB Decreased Osteoclast Differentiation of Primary-Cultured Monocytes

The imbalanced regulation of the bone-remodeling process promotes abnormal metabolic bone diseases, especially osteoporosis [1]. Osteoporosis, particularly for postmenopausal women, increases the bone resorption process because osteoclast differentiation occurs instead of bone formation, resulting in an enhanced risk of bone fragility and fractures [39]. To examine the effects of KB on bone resorption, this study investigated whether KB promoted the cellular differentiation of osteoclast cells isolated from the bone marrows of six-week-old mice. Monocytes from mouse bone marrow were successfully isolated and confirmed by a fluorescence-activated cell sorting (FACS) analysis with monocyte-specific surface markers (CD11b antibody) (Figure 4A). Monocytes are capable of osteoclast differentiation under a suitable microenvironment [40]. For the differentiation of primary-cultured monocytes to osteoclasts, cells were treated with macrophage colony-stimulating factor-1 (M-CSF) and RANKL. During a period of osteoclast differentiation, the treatment of KB did not affect the cell viability of primary-cultured monocytes of mouse bone marrow (Appendix A). After the induction of osteoclast differentiation of the primary-cultured monocytes, cells were treated with KB (5, 10, and 20 μM) for three and six days, and were analyzed by a TRAP activity assay and TRAP staining. The results showed that KB (10 and 20 μM) treatments significantly decreased TRAP activity and TRAP-positive staining colonies (Figure 4B,C and Appendix A). Next, we investigated the effects of KB on the expression of osteoclastic markers, such as tartrate-resistant acid phosphatase (*Trap*), nuclear factor of activated T cells cytoplasmic 1 (*Nfatc1*), and osteoclast-associated immunoglobulin-like receptor (*Oscar*). After treatment with KB (5, 10, and 20 μM), mRNA expressions of *Trap, Nfatc1*, and *Oscar* were assessed by qRT-PCR. KB significantly decreased the expressions of these osteoclast differentiation-related genes, compared to the non-treated control (Figure 4D). These results indicated that KB inhibited the osteoclast differentiation of monocytes from bone marrow.

### 2.5. KB Enhanced Osteoblast Differentiation in the Co-Culture System of MC3T3-E1 Cells and Primary-Cultured Monocytes

Bone remodeling is regulated by two processes: Bone formation by the differentiation of osteoblasts, and bone resorption by the differentiation of osteoclasts in the surface of the bone [32]. The relationship between bone formation and bone resorption includes the interaction of various cell types during bone remodeling [41,42]. Osteoclast and osteoblast cells communicate with each other through gap junctions [41]. The effects of KB on bone formation under the more closely physiological conditions of a co-culture system of preosteoblast MC3T3-E1 cells and primary-cultured monocytes were investigated. Previous studies suggested that a co-culture system containing both bone-forming cells (osteoblasts) and bone-resorbing cells (osteoclasts) more closely mimicked the in vivo environment as a cytocompatibility assessment in vitro for bone remodeling [43,44,45]. Consequently, a co-culture system of preosteoblasts and preosteoclasts was established. The osteoblast differentiation of co-cultured cells was induced by the treatment of ascorbic acid and β-glycerophosphate for three or six days, and the effects of co-treated KB (5, 10, and 20 μM) on osteoblast and osteoclast differentiations were examined. KB treatment increased ALP activity compared to the non-treated control cells, and a higher ALP activity was observed at three days of KB treatment compared to at six days (Figure 5A and Appendix A). However, KB did not influence TRAP activity in a co-cultured system at either three or six days of incubation (Figure 5B and Appendix A). 

The homeostasis of bone remodeling is maintained by the balance of bone resorption and bone formation [1]. A previous study demonstrated that a co-culture of preosteoblasts and preosteoclasts enhanced both osteoblast and osteoclast differentiation [43]. However, other studies have suggested that co-cultured cells enhanced osteoblast differentiation but reduced osteoclast differentiation [44,45]. In the present study, KB treatment promoted osteoblast differentiation but did not affect osteoclast differentiation in the co-culture system. Although KB reduced the osteoclast differentiation in primary-cultured monocytes (Figure 4), KB treatment in the co-culture system did not affect osteoclast differentiation. This may be due to the increased M-CSF and RANKL in the co-culture media. These results indicate that KB plays a key role in osteoblast differentiation during the bone remodeling process. 

### 2.6. KB Inhibited BMD Loss in OVX Mice

OVX mice present a reduced bone mass and quality, with significantly decreased BMD and bone mineral content (BMC) [46]. Based on the in vitro study, the anti-osteoporotic effects of KB in OVX mice were investigated. Seven-week-old female ddY mice underwent either an ovariectomy or sham-operated surgery (Sham). After surgery, the mice were divided into four groups: (1) Sham, (2) OVX control, (3) OVX treated with 2 mg/kg/day of KB, and (4) OVX administrated with 5 mg/kg/day of KB (*n* = 5 in each group). The treatments were orally administered to the mice for 12 weeks, and the BMD and BMC of the right femur were measured at 0, 6, and 12 weeks using a PIXI-mus bone densitometer (Figure 6A). At the end of the in vivo experiment, transverse microcomputed tomography (micro-CT) images of the right femur were scanned (Figure 6B), and the micro-CT images were analyzed for trabecular bone structural properties, including bone volume (BV/TV), trabecular thickness (Tb.Th), trabecular number (Tb.N), and trabecular spacing (Tb.Sp) (Figure 6C). 

As expected, the OVX mice showed a significantly reduced BMD in the right femur bone and impaired trabecular bone structural properties. The OVX group showed decreased BV/TV, Tb.Th, and Tb.N, and increased Tb.Sp compared to the Sham group, indicating that OVX-induced osteoporotic-bone loss was processed. However, BMC and food intake did not differ among the Sham, OVX control, or KB-administered groups. KB administration inhibited the OVX-induced BMD loss in the right femur bone and restored the impaired bone structural properties of BV/TV, Tb.Th, Tb.N, and Tb.Sp compared to the OVX control group (Figure 6). These results demonstrated the anti-osteoporotic effects of KB in vivo.

## 3. Materials and Methods

### 3.1. Fractionation, Isolation, and Structure Elucidation of the Bioactive Component

The dried LRC (1 kg) was extracted with 70% ethanol at room temperature, and the ethanol solution was then evaporated under reduced pressure. The 70% ethanol extract (97 g) was chromatographed on a Diaion HP-20 gel column and eluted with an H_2_O–Ethanol gradient system to obtain four fractions (A–D). Fraction B (11 g) showed that the most potent ALP activity was chromatographed on an RP-18 column (3 × 55 cm) and eluted with an H_2_O–methanol gradient (from 90:10 to 0:100, 800 mL for each step) to yield five fractions (B1–B5). Fraction B4 (98 mg) was further chromatographed on an RP-18 column (2 × 30 cm) and eluted with H_2_O–methanol (from 90:10 to 0:100, 400 ml for each step) to obtain a bioactive component (20 mg) (Appendix A). The structure of the bioactive component was elucidated by proton nuclear magnetic resonance (^1^H-NMR), carbon-13 nuclear magnetic resonance (^13^C-NMR), and mass spectrometry analyses (Appendix A), as well as by a comparison with the previously reported data [26] (Appendix A).

### 3.2. Cell Culture and Reagents

A mouse MC3T3-E1 cell line was purchased from the RIKEN Cell Bank (Tsukuba, Japan) and cultured in an α-modified minimal essential medium (α-MEM) supplemented with 10% fetal bovine serum, penicillin (100 U/mL), and streptomycin (100 μg/mL). Osteoblast differentiation was induced by adding an osteogenic medium containing ascorbic acid (50 ug/mL) and β-Glycerophosphate (10 mm) after allowing 24 hours for cell adherence (day 0). For the primary culture of monocytes, bone marrow cells were flushed from the femoral bones of six-week-old mice with a fine-bore syringe in the presence of 1 mm ascorbate-2-phosphate (Sigma-Aldrich; St. Louis, MO, USA). The isolated monocyte cells were identified by immunophenotypic analysis with a CD11b antibody (BioLegend; San Diego, CA, USA), using a FACS Aria III cell sorter (BD Biosciences; San Jose, CA, USA) and FACS Diva software (BD Biosciences). Monocyte cells were cultured in an α-MEM medium in the presence of 30 ng/mL of M-CSF (PeproTech; Rocky Hill, CT, USA) and 50 ng/mL of RANKL (PeproTech) for the induction of differentiation from preosteoclasts to osteoclasts [47]. For the co-culture system, MC3T3-E1 (2 ×1 0^4^ cells/well) cells and primary monocytes (4 × 10^4^ cells/well) were co-cultured in osteoblast differentiation media containing ascorbic acid (50 µg/mL) and β-glycerophosphate (10 mm). All cultured cells were incubated in a humidified atmosphere at 37 °C and 5% CO_2_. The cells were used at passages 5–10 after purchase for all experiments. The medium was changed every three days. KB was purchased from Aktin (Cat. No. APC-624) (Chengdu, China).

### 3.3. WST Assay and ALP Activity Assay and Staining

The primary-cultured monocytes and MC3T3-E1 cells (3 × 10^3^ cells/well) were incubated in a 96-well plate overnight and co-treated with different concentrations of KB (5, 10, and 20 μM). Cell viability was determined by a WST assay using an EZ-Cytox Cell Viability Assay Kit (Daeil; Seoul, Korea). A WST solution (20 μL, 5 mg/mL in phosphate-buffered saline) was added to each well, and the cells were incubated for another four hours. Absorbances were measured at 450 and 655 nm using a microplate reader (BIO-RAD; Hercules, CA, USA). Mouse MC3T3-E1 cells were lysed at 4 °C in a buffer containing 1 mmol/L Tris–HCl (pH 8.8), 0.5% Triton X-100, 10 mmol/L Mg^2+^, and 5 mmol/L *p*-nitrophenylphosphate as substrates. After homogenization, ALP activity was measured at the absorbance of 405 nm (BIO-RAD; Hercules, CA, USA). For ALP staining, cells were fixed in cold 4% paraformaldehyde for 10 min and washed with phosphate-buffered saline (PBS) three times. The fixed cells were stained with a BCIP/NBT (Sigma-Aldrich) for 30 minutes at room temperature. ALP-positive cells were determined under a light microscope.

### 3.4. Mineralized Nodule Formation in Osteoblast Cells

Preosteoblast MC3T3-E1 cells were cultured in a 48-well plate overnight and treated with 50 μg/mL of ascorbic acid and 10 mM of β-glycerophosphate with or without treatment with KB (10 and 20 μM) for three weeks. The cells were fixed with cold 70% ethanol for 10 min at room temperature and washed with water. Calcium deposits in the mineralized cells were determined by alizarin red S (Sigma-Aldrich) staining. Positive alizarin red S staining was determined using a light microscope. To quantify this, stained cells were extracted with 10% cetylpyridinium chloride for 1 h and transferred into a 96-well plate. The absorbance of the extracts was measured at a wavelength of 550 nm (BIO-RAD; Hercules, CA, USA).

### 3.5. Quantitative Reverse-Transcription PCR (qRT-PCR)

Total RNA was extracted from the cultured cells using the TRIzol reagent (Invitrogen; Carlsbad, CA, USA) following the manufacturer’s instructions, and the RNA quality was assessed by absorption measurement performed at 260/280 nm. The extracted RNA was subsequently reverse transcribed using a RevertAid™ H Minus First Strand cDNA Synthesis Kit (Fermentas; Hanover, NH, USA), with oligo(dT)_15–18_ as a random primer. All real-time reverse-transcription polymerase chain reaction (RT-PCR) measurements were performed using the ABI Prism 7000 Sequence Detection System (Applied Biosystems; Foster City, CA, USA). All PCR amplifications were performed in a total volume of 25 μL, containing 150 ng of cDNA using an SYBR Green I qPCR Kit (TaKaRa; Shiga, Japan) according to the manufacturer’s recommendations. The specific primers for the osteoblast markers were as follows: 5′-TCC CAC GTT TTC ACA TTC GG-3′ and 5′-GGC CAT CCT ATA TGG TAA CGG G-3′ for mouse *Alpl*, 5′-TAG TGA ACA GAC TCC GGC GCT A-3′ and 5′-ATG GCT TGA AGA CCG CCT ACA-3′ for mouse *Bglap,* 5′-ATG GCG TCC TCT CTG CTT G-3′ and 5′-TGA AAG GTC AGC GTA TGG CTT-3′ for mouse *Sp7*, 5′-TAC CTG TGT GGA CAT GAC C-3′ and 5′-CAG ATC CAT AGT GAA ACC GC-3′ for mouse *Trap,* 5′-GGA GAG TCC GAG AAT CGA GAT-3′ and 5′-TTG CAG CTA GGA AGT ACG TCT-3′ for mouse *Nfatc1,* 5′-CCT AGC CTC ATA CCC CCA G-3′ and 5′-CGT TGA TCC CAG GAG TCA CAA-3′ for mouse *Oscar,* and 5′-TGA CCA CAG TCC ATG CCA TC-3′ and 5′-GAC GGA CAC ATT GGG GGT AG-3′ for mouse *Gapdh*. By normalizing to *Gapdh*, the relative quantification of gene expression was performed using the comparative threshold (Ct) method, as described by the manufacturer (Applied Biosystems). The values were expressed as the fold change over the control. Relative gene expression was displayed as 2^-ΔCt^ (ΔCt = Ct _Target gene_ − Ct _Gapdh_). The fold change was calculated as 2^-ΔΔCt^ (ΔΔCt = ΔCt _Control_ − Ct _Treatment_).

### 3.6. Co-Culture System of MC3T3-E1 Cells and Primary Monocytes 

For the co-culture system, MC3T3-E1 (2 × 10^4^ cells/well) cells were cultured in a 48-well plate overnight, and the isolated monocytes (4 × 10^4^ cells/well) from the mouse bone marrow were added to the MC3T3-E1 cells and incubated for one day. The co-culture of MC3T3-E1 cells and primary-cultured monocyte cells was co-incubated with an osteoblast differentiation–induction media containing ascorbic acid (50 µg/mL) and β-glycerophosphate (10 mM) with or without treatment with KB (10 and 20 μM) for either three or six days. The differentiations between osteoblasts and osteoclasts were measured by ALP activity and TRAP activity, respectively.

### 3.7. TRAP Staining and Activity Test

After the induction of the primary-cultured monocytes, the medium was removed and gently washed with PBS. The differentiated osteoclast cells from the monocytes were measured by a TRAP activity assay and were further stained using an Acid-Phosphatase Kit (Sigma-Aldrich). The absorbance was measured at 405 nm (BIO-RAD; Hercules, CA, USA), and TRAP activity was expressed as the percentage of the untreated control. For TRAP staining, the cells were fixed in cold 4% paraformaldehyde for 10 min and washed with PBS. The fixed cells were stained with an Acid Phosphatase and Leukocyte Kit (Sigma-Aldrich) for two hours at room temperature. TRAP-positive multinucleated cells containing three or more nuclei were counted under a light microscope.

### 3.8. Ovariectomized Osteoporosis Model Mice

The OVX and sham-operated eight-week-old female ddY mice were purchased from Shizuoka Laboratory Center Inc. (Hamamatsu, Japan). The mice were acclimated for 10 days prior to experimentation. The mice were maintained on a diet (5.0 g/day) of Formula-M07 (Feedlab Co. Ltd.; Hanam, Korea) and tap water (15 ml/day). All mice were housed individually in clear plastic cages under controlled temperature (23 ± 2 °C), humidity (55 ± 5%), and illumination (12-hour light/dark cycle). Mice were orally administered KB (2 or 5 mg/kg/day) for 12 weeks. The BMD and BMC were measured at 0, 6, and 12 weeks during the experiment. The animal research procedures were approved by the Animal Care and Use Committee of the Ajou University School of Medicine (IACUC No. 2014-0066, approved at 23 September 2015), and all experiments were conducted in accordance with the institutional guidelines established by the committee. All efforts were made to minimize animal suffering and to reduce the number of mice used.

### 3.9. Measurement of BMD and Micro-CT Imaging in Bone Tissues

After anesthetization using tiletamine/zolazepam (Zoletil; Virbac Laboratories, Carros, France), the mice were placed on the specimen tray for measurements. The BMC and BMD of the right femur were measured using a PIXI-mus bone densitometer with on-board PIXI-mus software (GE Lunar; Madison, WI, USA) for small animals, which was adjusted in relation to body weight. Transverse micro-CT images of the right femur were scanned using a high-energy spiral scan micro-CT, Skyscan 1173 (Bruker; Kontich, Belgium), which had a voltage of 60 kV, a current of 400 μA, an exposure time of 400 ms, a charge coupled device (CCD) camera readout of 1280 × 1280, and rotation steps of 360. Reconstruction was performed using NRecon software (Bruker microCT). Two-dimensional axial and three-dimensional images were reconstructed for qualitative and quantitative analyses. Axial images were displayed using Inveon Research Workplace (Siemens) with a spatial resolution of 8.88 μm. Axial reformats were performed to allow slice-by-slice, manual tracing of the contours of the trabecular bone. For the cross-sectional study of the trabecular bone, the region of interest (ROI) measurements of 1500 μm trabecular bone were analyzed approximately 300 μm below the growth plate. The analysis provided information about the main histomorphometric parameters, including bone volume (BV/TV, %), trabecular number (Tb.N, 1/mm), thickness (Tb.Th, mm), and spacing (Tb.Sp, mm). 

### 3.10. Statistical Analysis

All the experiments were repeated at least three times unless stated otherwise, and the results were presented as the means ± standard deviations, as indicated. A statistical software package (SPSS 11.0 for Windows, SPSS Inc. Chicago, IL, USA) was used to perform the statistical tests. The statistical significance of the differences between groups was assessed by the Student’s t-test. Comparisons of multiple groups were done with a one-way analysis of variance (ANOVA), followed by Tukey’s HSD (honest significant difference) post hoc test for correction of multiple comparisons. A probability value (*p*) less than 0.05 (*p* < 0.05) was considered statistically significant. The results were expressed as mean ± SEM.

## 4. Conclusions

This is the first study to evaluate the anti-osteoporotic effects of kukoamine B (KB) isolated from LRC in vitro and in vivo. The in vitro experiments demonstrated that KB enhanced the differentiation and mineralized nodule formation of preosteoblastic MC3T3-E1 cells, and inhibited the osteoclast differentiation of preosteoclast-lineage monocytes isolated from mice bone marrow. In a co-culture of MC3T3-E1 cells and primary-cultured monocytes, KB treatment promoted osteoblast differentiation, but not osteoclast differentiation. The in vivo experiments demonstrated that KB significantly improved OVX-induced bone mineral density loss and impaired bone structural properties in osteoporosis model mice. The results suggest that KB may be a good therapeutic candidate for the treatment and prevention of osteoporosis.

## Figures and Tables

**Figure 1 ijms-20-02784-f001:**
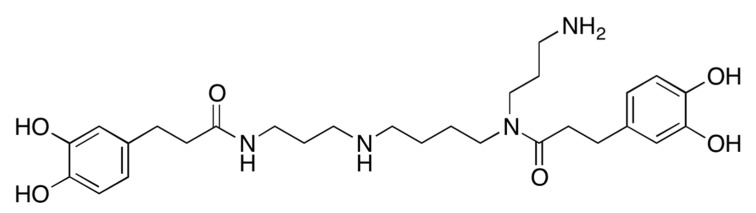
The chemical structure of the isolated kukoamine B.

**Figure 2 ijms-20-02784-f002:**
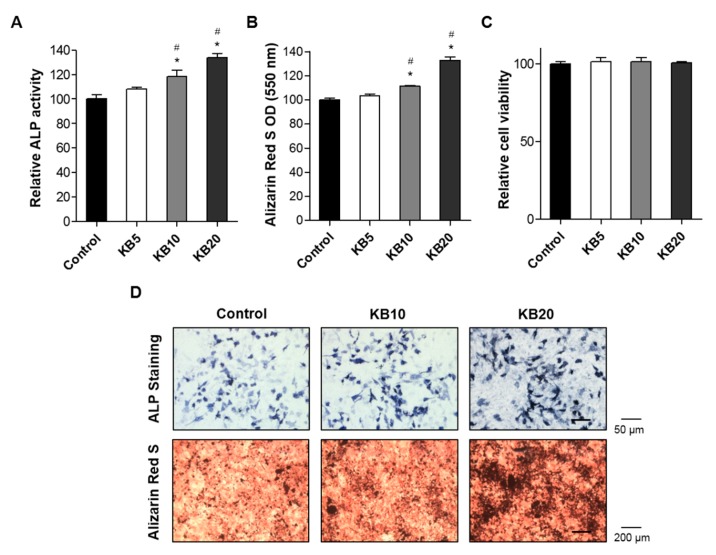
The effects of kukoamine B (KB) on cellular differentiation, cell viability, and mineralized nodule formation of the preosteoblast MC3T3-E1 cells. (**A**,**B**) Assessment of alkaline phosphatase (ALP) activity and alizarin red S optical density (OD) value (550 nm) in KB-treated MC3T3-E1 cells. After induction of osteoblast differentiation with 50 μg/mL of ascorbic acid and 10 mM of β-glycerophosphate, cells were treated with three different KB concentrations (5, 10, and 20 µM) for 3 and 21 days, respectively, and then ALP activity and alizarin Red S OD value were assessed. *: *p* < 0.05 vs. Control, ^#^: *p* < 0.05 vs. KB5 (Tukey’s honest significant difference (HSD) post hoc test, ANOVA). (**C**) Assessment of the cell viability in the KB-treated MC3T3-E1 cells. Cells were treated with three different concentrations of KB (5, 10, and 20 µM) for three days, and then cell viability was assessed. (**D**) Assessment of ALP staining and in vitro bone mineralization in the KB-treated MC3T3-E1 cells. After the induction of osteoblast differentiation, the cells were treated with 10 and 20 µM of KB for three days (for ALP staining) or 21 days (for mineralized nodule formation staining), and then cells were stained with ALP and alizarin red S. The positively stained cells and nodules were visualized under a microscope. Control: KB non-treated cells.

**Figure 3 ijms-20-02784-f003:**
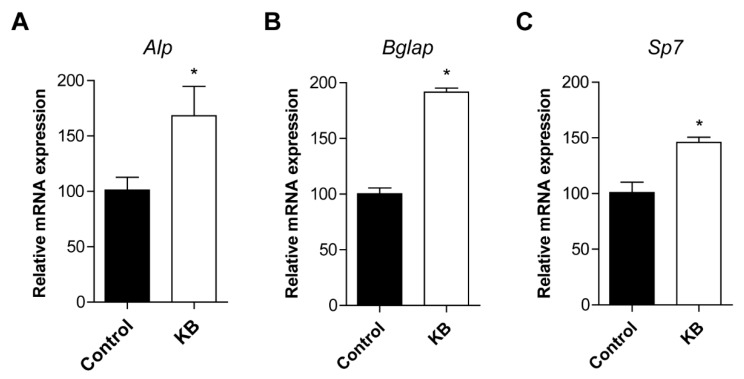
The effects of kukoamine B (KB) on the mRNA expression levels of osteoblastic markers in MC3T3-E1 cells. After the induction of osteoblast differentiation, the cells were treated with 20 μM of KB. After treatment, the total RNA of the cells was extracted and the mRNA expression levels of *Alpl* (alkaline phosphatase, ALP) (**A**), *Bglap* (bone gamma-carboxyglutamate protein, Osteocalcin) (**B**), and *Sp7* (Osterix) (**C**) genes were assessed by quantitative reverse-transcription polymerase chain reaction (qRT-PCR). The mRNA levels of the osteoblastic markers were normalized by *Gapdh* (glyceraldehyde 3-phosphate dehydrogenase) mRNA expression. Control: Non-KB-treated cells. *: *p* < 0.05 vs. Control (Student’s t-test). All experiments were repeated three times.

**Figure 4 ijms-20-02784-f004:**
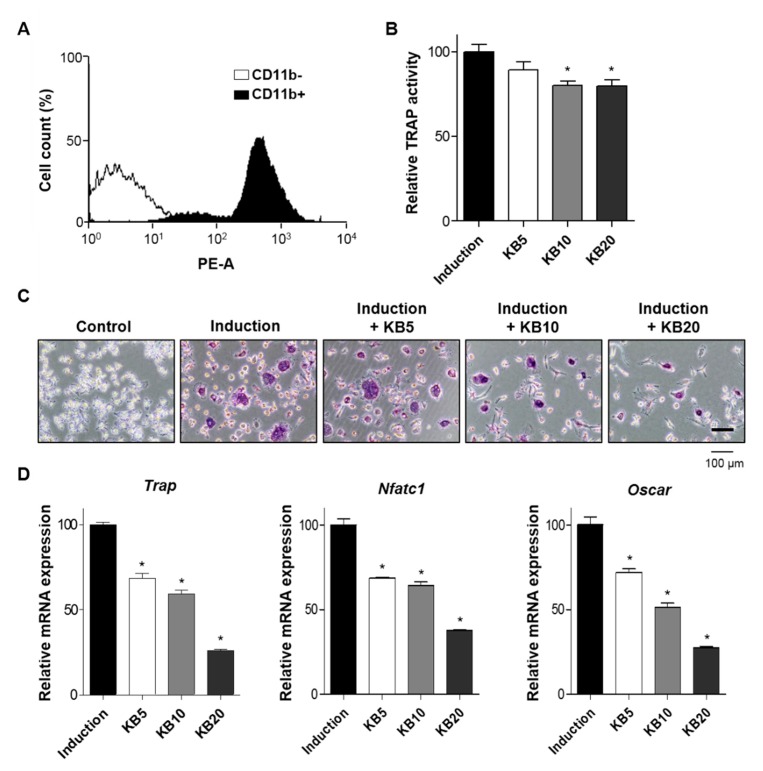
The effects of kukoamine B (KB) on osteoclast differentiation of primary-cultured monocytes. (**A**) The validation of the successful isolation of monocytes from mouse bone marrow. Primary-cultured monocytes were identified by an immunophenotypic analysis with a monocyte-specific surface positive marker (phycoerythrin-conjugated CD11b antibody) using a fluorescence-activated cell sorting analysis. (**B**,**C**) The assessment of tartrate-resistant acid phosphatase (TRAP) activity in the KB-treated monocyte cells. After the induction of osteoclast differentiation, the cells were treated with KB (5, 10, and 20 µM) for six days, and then TRAP activity was assessed (**B**). The cells were also stained with a TRAP staining kit, and the differentiated osteoclast cells were visualized under a microscope (**C**). Control: Non-induction of osteoclast differentiation. Induction: Induction of osteoclast differentiation with 30 ng/mL of a macrophage colony-stimulating factor (M-CSF) and 50 ng/mL of a receptor activator of nuclear factor kappa-B ligand (RANKL). (**D**) The mRNA expression levels of the tartrate-resistant acid phosphatase (*Trap*), nuclear factor of activated T-cells cytoplasmic 1 (*Nfatc1*), and osteoclast-associated immunoglobulin-like receptor (*Oscar*) genes were measured by qRT-PCR. All experiments were repeated three times. *: *p* < 0.05 vs. Induction (Tukey’s HSD post hoc test, ANOVA).

**Figure 5 ijms-20-02784-f005:**
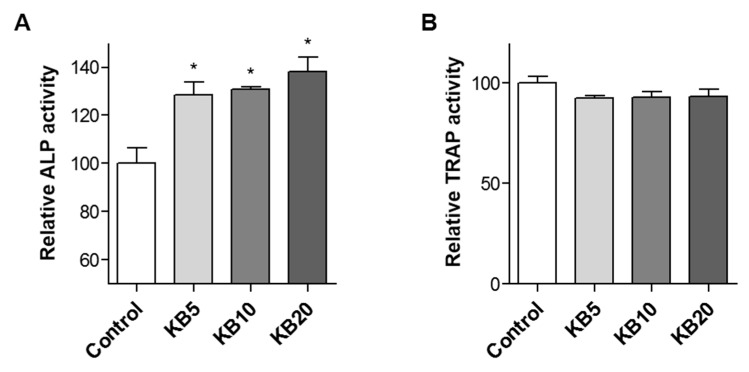
The effects of kukoamine B (KB) on osteoblast and osteoclast differentiation in the co-culture of preosteoblasts and primary-cultured monocytes. Co-cultured MC3T3-E1 and primary-cultured monocyte cells were treated with osteoblast differentiation reagents ascorbic acid and β-glycerophosphate, and then co-treated with KB (5, 10, and 20 μM) for three days. Alkaline phosphatase (ALP) activity (**A**) and tartrate-resistant acid phosphatase (TRAP) activity (**B**) were assessed in the co-culture cells. Control: KB non-treated cells. *: *p* < 0.05 vs. Induction (Tukey’s HSD post hoc test, ANOVA).

**Figure 6 ijms-20-02784-f006:**
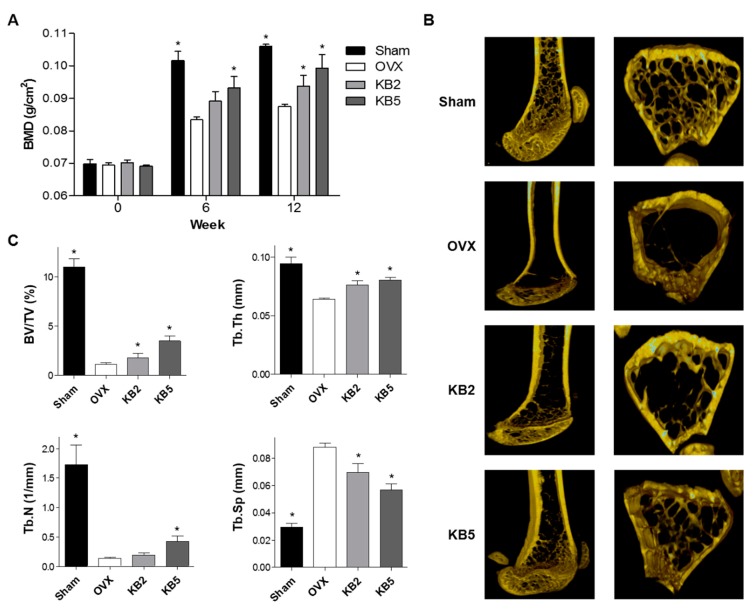
The effects of kukoamine B (KB) on the improvement of bone mineral density (BMD) and bone structural properties in ovariectomized (OVX) osteoporosis model mice. The OVX mice were administered with KB (2 or 5 mg/kg/day) for 12 weeks. Sham: Sham-operated group (*n* = 5), OVX: KB non-administered mice group (*n* = 5). (**A**) BMD of the right femur was measured using a PIXI-mus bone densitometer at 0, 6, and 12 weeks. (**B**) Transverse microcomputed tomography (micro-CT) images of the right femur were scanned at the end of the experiment. (**C**) Trabecular bone structural properties, including bone volume (BV/TV), trabecular thickness (Tb.Th), number (Tb.N), and spacing (Tb.Sp), were analyzed at the end of the experiment. *: *p* < 0.05 vs. OVX group.

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
