# Peer review of "Anti-Osteoporotic Effects of Kukoamine B Isolated from Lycii Radicis Cortex Extract on Osteoblast and Osteoclast Cells and Ovariectomized Osteoporosis Model Mice"

_ijms, 2019, doi:10.3390/ijms20112784_

Reviewer 1 Report

The manuscript was substantially improved. I do not have any concerns. 

Reviewer 2 Report

Comments are incorporated in revised version.

This manuscript is a resubmission of an earlier submission. The following is a list of the peer review reports and author responses from that submission.

Round  1

Reviewer 1 Report

The current work is interesting and technically sound. However, there are certain concerns that need to be addressed:

1.      Title should be more precise and clearer.

2.      Provide functional assay for osteoclasts upon Kukoamine B treatment such as bone resorption assay using calcium phosphate coated surfaces or bone slices.

3.      Biomarker analysis (anabolic or catabolic) in serum.

4.      Since osteoclasts are osteoblasts have a coupling effect what do you expect if condition media from WT osteoclasts is added to cultured osteoblasts (WT and KB treated).   Do you expect any changes in activity of the osteoblasts?

5.      Provide clear TRAP images of osteoclasts treated with KB.

Author Response

Manuscript ID (IJMS-485974)

Thank you for your valuable comments on our manuscript. We have revised the manuscript according to your comments and supplied detailed point-by-point responses below. All changes in the text appear in red. The English language of the manuscript was proofread and edited by native English-speaking professional scientists at EDITAGE, a scientific research paper editing company.

Reviewer 1

The current work is interesting and technically sound. However, there are certain concerns that need to be addressed:

1.  Title should be more precise and clearer.

As per the reviewer’s comments, the title was changed to “Anti-osteoporotic effects of kukoamine B isolated from Lycii radicis cortex extract on osteoblast and osteoclast cells and ovariectomized osteoporosis model mice”.

2.  Provide functional assay for osteoclasts upon Kukoamine B treatment such as bone resorption assay using calcium phosphate coated surfaces or bone slices.

As soon as we received the reviewer’s comments, we ordered a bone resorption assay kit (CSR-BRA-48KIT) and expected it to arrive at our lab within 2~3 weeks. However, we are still awaiting the delivery of the kit. As a result, we could not perform the bone resorption assay before the submission deadline of these revisions. We would carry out the bone resorption assay when we receive the kit and provide the results to you as soon as possible.

3.  Biomarker analysis (anabolic or catabolic) in serum.

We agree with the reviewer’s suggestion regarding the biomarker analysis (anabolic or catabolic) in serum. Unfortunately, we did not collect blood samples from mice at the end of animal experiment.

4. Since osteoclasts are osteoblasts have a coupling effect what do you expect if condition media from WT osteoclasts is added to cultured osteoblasts (WT and KB treated). Do you expect any changes in activity of the osteoblasts?

Based on the reviewer’s comments, we added the condition media from WT osteoclasts culture to WT and KB-treated osteoblasts. There was no significant difference in osteoblast differentiation between WT and KB-treated osteoblast cells.

5. Provide clear TRAP images of osteoclasts treated with KB.

According to the reviewer’s comments, we have provided clear TRAP staining images in Figure 4C.

Reviewer 2 Report

The authors identified Kukoamine B, a bioactive compound fractionated from Lycii Radicis Cortex extract, as an anti-osteoporotic component. Although some of the in vitro data are valid, this manuscript is basically incomplete. Plenty of experiments need to be done to meet their conclusion. At some extent, their results can not support their conclusion. Overall, both the integrity and quality of this manuscript must be improved before publication.

Major concerns:

1.  The authors aimed to screen the bioactive fraction responsible for anti-osteoporotic effects in LRC extract. They need to show all of the compounds that were examined in the ALP assay.

2.  WST assay is an indicator of cell viability but not an indicator of cell proliferation. To evaluate proliferation rate, EdU/BrdU assay need to be done.

3.  In Fig 2c, quantification of Alizarin red staining is required.

4.  In Fig 4c, there is basically no mature osteoclasts marked by actin-ring in the TRAP staining images. I am not sure if the differentiation system is not working. Quantifications of osteoclasts number and nucleus number are necessary. Gene expression level of osteoclast markers is also necessary.

5.  In Fig 5, the co-culture system here is meaningless. First, the authors should culture primary calvarial osteoblasts  with primary bone morrow macrophage, but not pre-osteoblast cell line. Second, to determine the cross-talk between osteoblasts and osteoclasts, Vitamin D3 and PGE2 should be added into the system to induce the production of M-CSF and RANKL, but not the osteogenic medium. Third, the primary monocytes can not even survive without M-CSF.

6.  In Fig 6, both static and dynamic histomorphometry are required.

Minor concern:

1.  Overall the language is confusing and riddled with grammatical errors and typos. Having a native English speaker edit this manuscript would be helpful.

Author Response

Manuscript ID (IJMS-485974)

Thank you for your valuable comments on our manuscript. We have revised the manuscript according to your comments and supplied detailed point-by-point responses below. All changes in the text appear in red. The English language of the manuscript was proofread and edited by native English-speaking professional scientists at EDITAGE, a scientific research paper editing company.

Reviewer 2

The authors identified Kukoamine B, a bioactive compound fractionated from Lycii Radicis Cortex extract, as an anti-osteoporotic component. Although some of the in vitro data are valid, this manuscript is basically incomplete. Plenty of experiments need to be done to meet their conclusion. At some extent, their results cannot support their conclusion. Overall, both the integrity and quality of this manuscript must be improved before publication.

Major concerns:

1.  The authors aimed to screen the bioactive fraction responsible for anti-osteoporotic effects in LRC extract. They need to show all of the compounds that were examined in the ALP assay.

As per the reviewer’s comments, we have included the ALP activity assay results of all fractions isolated from the LRC extract in Fig S2.

2.  WST assay is an indicator of cell viability but not an indicator of cell proliferation. To evaluate proliferation rate, EdU/BrdU assay need to be done.

Based on the reviewer’s comments, we have replaced the term “cell proliferation” with “cell viability”.

3.  In Fig 2c, quantification of Alizarin red staining is required.

As per the reviewer’s comments, the quantification of Alizarin red staining has been included in Figure 2B.

4.  In Fig 4c, there is basically no mature osteoclasts marked by actin-ring in the TRAP staining images. I am not sure if the differentiation system is not working. Quantifications of osteoclasts number and nucleus number are necessary. Gene expression level of osteoclast markers is also necessary.

As per the reviewer’s comments, we have re-examined the osteoclast differentiation in order to obtain clear TRAP staining images presented in Fig 4C. The gene expression levels of osteoclast differentiation-related genes were measured by quantitative RT-PCR and are illustrated in Fig 4D.

5.  In Fig 5, the co-culture system here is meaningless. First, the authors should culture primary calvarial osteoblasts with primary bone morrow macrophage, but not pre-osteoblast cell line. Second, to determine the cross-talk between osteoblasts and osteoclasts, Vitamin D3 and PGE2 should be added into the system to induce the production of M-CSF and RANKL, but not the osteogenic medium. Third, the primary monocytes cannot even survive without M-CSF.

We have added a more detailed explanation of the co-culture experiments. We quantified the ALP and TRAP activities using the same protocol used in the single-culture system (References: Chen, S.; Ye, X.; Yu, X.; Xu, Q.; Pan, K.; Lu, S.; Yang, P., Co-culture with periodontal ligament stem cells enhanced osteoblastic differentiation of MC3T3-E1 cells and osteoclastic differentiation of RAW264.7 cells. Int J Clin Exp Pathol 2015, 8, (11), 14596-607; Wu, L.; Feyerabend, F.; Schilling, A. F.; Willumeit-Romer, R.; Luthringer, B. J., Effects of extracellular magnesium extract on the proliferation and differentiation of human osteoblasts and osteoclasts in coculture. Acta Biomater 2015, 27, 294-304; Bernhardt, A.; Thieme, S.; Domaschke, H.; Springer, A.; Rosen-Wolff, A.; Gelinsky, M., Crosstalk of osteoblast and osteoclast precursors on mineralized collagen towards an in vitro model for bone remodeling. J Biomed Mater Res A 2010, 95, (3), 848-56).

In our pre-osteoblast monocyte culture system, no primary monocyte death occurred. This is probably due to the production of M-CSF by osteoblast cells (Reference: Yao, G.Q.; Sun, B.H.; Weir, E.C.; Insogna, K.L., A role for cell-surface CSF-1 in osteoblast-mediated osteoclastogenesis. Calcif Tissue Int 2002, 70, (4), 339-46).

KB treatment promoted osteoblast differentiation but did not affect osteoclast differentiation in the co-culture system. Although KB reduced osteoclast differentiation in single-culture of monocytes, no change in osteoclast differentiation was found when KB was treated under the co-culture system. This may be due to increased M-CSF and RANKL in the co-culture media. These results indicated that KB plays a key role in osteoblast differentiation during the bone remodeling process.

6.  In Fig 6, both static and dynamic histomorphometry are required.

  Thank you for your valuable comments. We found that the right femur bone samples which were previously used for acquiring the transverse microcomputed tomography (micro-CT) images and for measuring BMD were dried up and contaminated. Therefore, we could not perform both static and dynamic histomorphometry.

Minor concern:

1.  Overall the language is confusing and riddled with grammatical errors and typos. Having a native English speaker edit this manuscript would be helpful.

 → The English language of the manuscript has now been proofread and edited by native English-speaking professional scientists at EDITAGE, a scientific research paper editing company.
